

# Fast cyclic stimulus flashing modulates perception of bi-stable figure

Henrikas Vaitkevicius[1], Vygandas Vanagas[1], Alvydas Soliunas[2], Algimantas Svegzda[1], Remigijus Bliumas[1], Rytis Stanikunas[1] and Janus J. Kulikowski[3]

[1] Institute of Psychology, Vilnius University, Vilnius, Lithuania
[2] Institute of Biosciences, Vilnius University, Vilnius, Lithuania
[3] Faculty of Life Sciences, University of Manchester, Manchester, United Kingdom

## ABSTRACT

Many experiments have demonstrated that the rhythms in the brain influence the initial perceptual information processing. We investigated whether the alternation rate of the perception of a Necker cube depends on the frequency and duration of a flashing Necker cube. We hypothesize that synchronization between the external rhythm of a flashing stimulus and the internal rhythm of neuronal processing should change the alternation rate of a Necker cube. Knowing how a flickering stimulus with a given frequency and duration affects the alternation rate of bistable perception, we could estimate the frequency of the internal neuronal processing. Our results show that the perception time of the dominant stimulus depends on the frequency or duration of the flashing stimuli. The duration of the stimuli, at which the duration of the perceived image was maximal, was repeated periodically at 4 ms intervals. We suppose that such results could be explained by the existence of an internal rhythm of 125 cycles/s for bistable visual perception. We can also suppose that it is not the stimulus duration but the precise timing of the moments of switching on of external stimuli to match the internal stimuli which explains our experimental results. Similarity between the effects of flashing frequency on alternation rate of stimuli perception in present and previously performed experiment on binocular rivalry support the existence of a common mechanism for binocular rivalry and monocular perception of ambiguous figures.

Corresponding author
Alvydas Soliunas,
alvydas.soliunas@gf.vu.lt

## INTRODUCTION

Neurophysiological studies addressing the coding of visual information in the brain led to the discovery of neurons which respond selectively to specific features of visual stimulus such as size, color, orientation, movement, characteristics of contour and spatial location, subsequently called feature detectors (*Lettvin et al., 1959*; *Hubel & Wiesel, 1959*; *Barlow, 1972*; *Barlow, Blakemore & Pettigrew, 1967*; *DeValois, 1973*; *Bishop, 1996*; *Bishop & Pettigrew, 1986*). It was believed that the responses of these detectors lead to the identification of various features of stimuli essential for recognition. However, the detecting of only physical properties of stimuli is not sufficient to explain all phenomena of stimulus perception. In some situations, internally formed features are not unequivocal. In

bistable perception, when two different sets of features are formed from the same physical stimulus, two different percepts result. For example, if the Necker cube is presented, one 3D feature set will represent one spatial orientation of the Necker cube and another 3D feature set will represent a different cube orientation. However, the physical stimulus on the retina remains unchanged. As was shown in the Leopold experiments (*Leopold et al., 2002*; *Leopold & Logothetis, 1999*), the perception time of a dominant image increases when a bistable stimulus is switched off for a few seconds and is switched on again. It can be assumed that at least two processes must be distinguished in this case: one related to the maintenance of the dominant image, another related to the influence on the alteration of the perceived image. For perceptual alteration to occur, one should have an alternative. However, after the stimulus is switched off, the subject sees nothing (there is no alternative stimulus in iconic memory) (*Leopold & Logothetis, 1999*). A trace of the dominant image is maintained in the "top-down" streams only but not in iconic memory. In this case, the absence of a stimulus on the input of the sensory system can extend the perception time of the dominant image. Such findings demonstrate that the process related to the alternation of percepts operates with two streams of information: a "bottom-up" stream from sensory input and a "top-down" stream from areas of the brain where different stimulus feature sets are represented.

Other evidence of interaction between the two mentioned streams was shown in *Stanley et al. (2012)* and other studies. It has been shown that when the external rhythm of a stimulus presentation coincided with the internal high-frequency rhythm, the signal dispersion decreased and its effectiveness increased (*Montemurro et al., 2008*; *Fründ et al., 2008*; *Cardin et al., 2009*; *Siegel, Warden & Miller, 2008*; *Vinck et al., 2010*; *Stanley et al., 2012*). *Stanley et al. (2012)* showed that the selectivity of cat LGN neurons, sensitive to the direction of movement and orientation, increased with the occurrence of synchronous firing (external and internal streams) at the inputs of neurons. These authors proposed that the only summation of excitation signals coming through different channels could not explain the observed effect. In this case the precise timing of incoming spikes but not the total cumulative effect is important. Moreover, it has been demonstrated that the greater the phase shift in time between the two sequences of spikes affecting inputs of a neuron, the less the cumulative influence of these streams of impulses on neuron activity (*Bi & Poo, 2001*; *Zhang et al., 1998*; *Song, Miller & Abbott, 2000*).

The neural rhythms are investigated by various neurophysiological and brain imaging methods. We cannot investigate these rhythms directly by psychophysical methods, but we can use the process of synchronization of two rhythms: internal neural rhythm and rhythm of external stimulation. We can present some visual stimuli rhythmically with some frequency and we could expect that it will generate internal bottom-up rhythms in peripheral parts of the neural circuitry that interact with the more central inherited rhythms of a particular frequency. As a result of such interactions, we could expect a change of perception's accuracy or response time depending on the frequency of stimulation. In other words, changing the frequency of external stimulation we can catch the frequency of an internal rhythm that is responsible for a particular perceptual process. This has been done by us (*Geissler et al., 2012*) in a study of binocular rivalry, where orthogonal black bars were

presented for the left and right eyes separately. Bars periodically flashed with a frequency of 25–125 flashes/s. It was found that the duration of the dominant time as a function of flashing frequency changed periodically, with the period about 4–5 ms. The present study is a continuation of previous work but now we investigated another phenomenon of bistable perception—the perception of ambiguous figures. The question is whether there is a common mechanism responsible for different types of bistable perception: binocular rivalry and perception of ambiguous figures.

## METHODS

### Participants

Eight subjects (seven males, one female) participated in the experiments. Three subjects (**RB**, **AS**, and **MK;** 32, 66, and 22 years old, respectively) had experience in psychophysical research. The other five (**AV**, **IS**, **GS**, **MR,** and **AS2;** 19, 20, 21, 22, and 49 years old, respectively) were naive. Two experienced subjects participated in 20 sessions, one experienced and one naive subject participated in 10 sessions and the remaining four participated in only three sessions. The results from the first session for one subject who participated in 20 sessions were removed from the data analysis because they varied significantly and differed significantly from the rest of his sessions. Subjects participated in one session per day, every day or with a less than 5-day interval. All subjects signed an informed consent approved by Vilnius Region Ethics Committee of Biomedical Research (approval No. 158200-13-578-173).

### Stimuli and apparatus

The stimulus was the standard Necker cube, drawn in black lines on a white (85 cd/m2—measured with PR680) background. The stimulus size was 1.7 × 1.7 degrees of visual angle. The transparent slide with the Necker cube was mounted on the specially designed tachistoscope with a 20 mm aperture. It had a chin or headrest, stabilizing the subject's head. A white PC-controlled LED illuminated the slide; a specially written program controlled an electric circuit to form the LED luminous flux and the data was transmitted to a PC through an LPT port. The structural diagram of the experimental apparatus is presented in Fig. 1A. The stimulus was switched on and off, i.e., flashed rhythmically at selected frequencies. The flash duration had an accuracy of about 5 µs. Subjects watched the stimulus monocularly with the right eye and this flashing image was seen for a fixed period. Subjects responded by pressing a key on a response box connected to the LPT port of the PC.

### Procedure

The experiments were carried out in a dark and partly soundproofed room. Before each session, the subject adapted to darkness for 10 min. The Necker cube was rhythmically turned on and off within a block lasting 180 s (Fig. 1B). On and off periods were equal and the frequency (and flash duration) of presentation in one block was constant. Eighteen blocks, each with a different flash duration (which we refer to as "stimulus duration") and with a 60 s pause in between the blocks, were randomly presented during each session. The minimum flash duration was 4 ms, the maximum 20 ms, i.e., flash duration varied

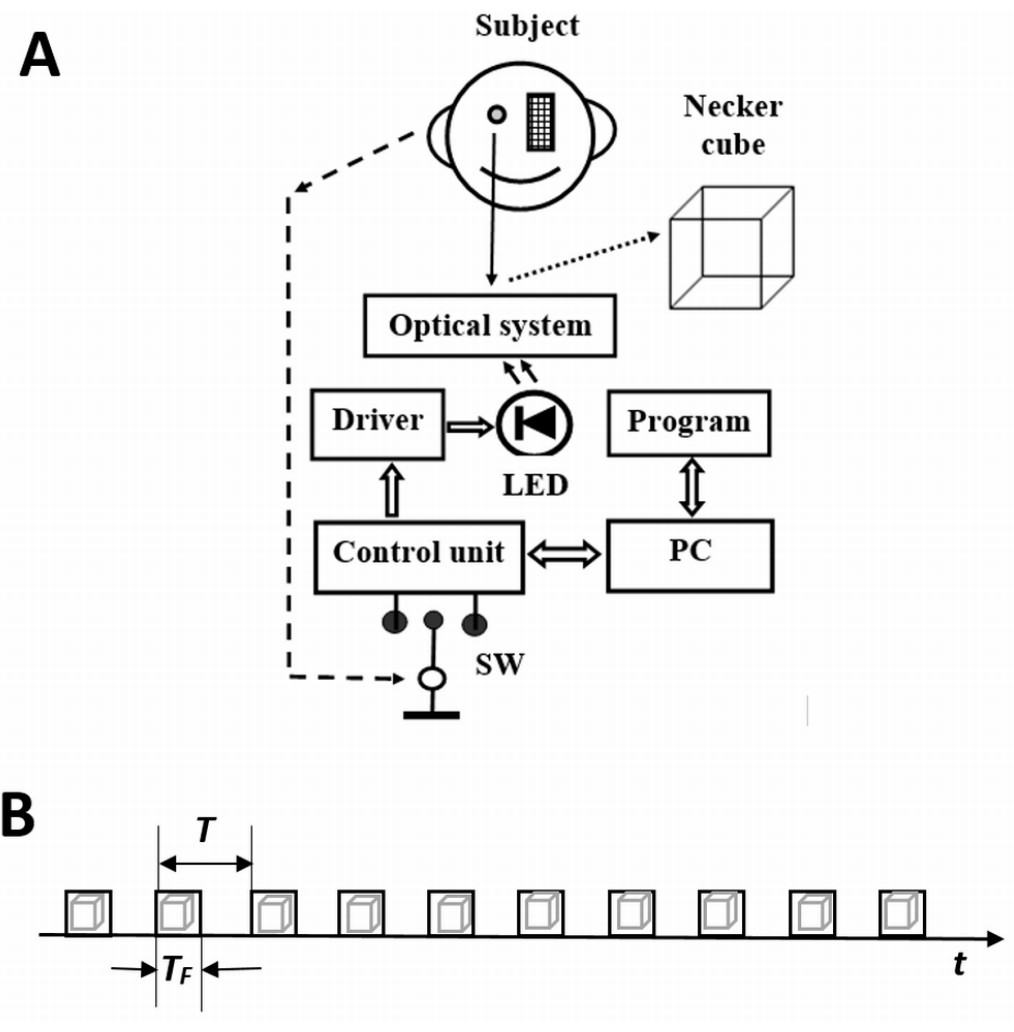

**Figure 1** **(A) Structural diagram of the experimental apparatus and (B) stimulus presentation procedure within one block.** LED, white light diode (color coordinates $x = 0,3262$, $y = 0,3351$); SW, switch (response key), PC, personal computer, Optical system, transparent slide with the Necker cube visible through 20 mm aperture. $T_F$, flash duration; $T = 2T_p$, interval between two flashes.

in 1 ms steps among the different blocks. The non-flashing Necker cube was presented in one block. The position of the Necker cube was called the "up position" (abbreviated UP) if its front wall was perceived higher than the rear, and the alternative position was called the "down position" (abbreviated DOWN). The task was to press and keep the key pressed when the position of the Necker cube DOWN was perceived and to release and keep the key released when the position of the Necker cube UP was perceived. Knowing the moments of perceptual changes, it was possible to determine the duration of perception of each of the Necker cube positions. We named this duration of perception an absolute perception time (abbreviated PT).

## Data analysis

The analysis of the obtained data was carried out with the same analysis methods as described in the paper of *Geissler et al. (2012)*. Firstly, as the perception time (PT) of UP and DOWN orientations of the Necker cube varied between subjects and sessions, instead of PT, we analyzed the deviations of PT (abbreviated dPT) from mean PT (i.e., absolute differences between each value of PT and the mean PT in an experimental session): i.e., $\Delta\tau_{\text{UP}}(k,i) = \bar{\tau}_{\text{UP}}(i) - \tau_{\text{UP}}(k,i)$ and $\Delta\tau_{\text{DOWN}}(k,i) = \bar{\tau}_{\text{DOWN}}(i) - \tau_{\text{DOWN}}(k,i)$. Here $i$ = the number of session, $k$ = the block number ($k = 1, \ldots, 18$), $\bar{\tau}_{\text{UP}}(i)$ and $\bar{\tau}_{\text{DOWN}}(i)$ is the mean PT of respectively UP and DOWN for all chosen blocks $k$ over whole $i$th session, and $\tau_{\text{UP}}(k,i)$ and $\tau_{\text{DOWN}}(k,i)$ is the mean dPT of UP and DOWN for block $k$ of the $i$th session. Next, we calculated the means of dPT for the $k$-th blocks over all $n$ sessions separately for UP and DOWN, i.e., $\Delta\tau_{\text{UP,DOWN}}(k) = \sum_{i=1}^{n} \Delta\tau_{\text{UP,DOWN}}(k,i)/n$, and joint averaged UP and DOWN function: $\Delta\tau(k) = M(\Delta\tau_{\text{UP}}(k) + \Delta\tau_{\text{DOWN}}(k))$.

Secondly, in order to check whether the dominant time significantly depends on the frequency of the flashing stimuli, one factor ANOVA was run for each subject separately.

Next, the PCA (principal component analysis) was run on the set of data (18 means of blocks and $n$ sessions). The main purpose of the PCA was to find how many factors influence the perception of ambiguous figures. In a similar investigation on binocular rivalry (*Geissler et al., 2012*; *Blake & Lee, 2005*), it was shown that the duration of the dominant time of the perceived stimuli depends on many factors. As different observers could produce different function of perception times on the frequency of stimulus, it is more informative to run the PCA separately on the data of different observers, however, the data of only three subjects (**RB**, **AS**, and **MK**) were sufficient for the PCA. Additionally, the data of **AV**, **IS**, **GS**, **MR** and **AS2** subjects (abbreviated **Rm5**) was aggregated. Such a choice let us to compare the averaged function for five subjects with the function for separate subjects.

As not all extracted factors may be significant (some may be related to random changes/fluctuations), we need to identify non-random factors. One of the most commonly used methods is the Kaiser's criterion (*Fabrigar et al., 1999*), which retains factors with eigenvalues greater than 1. It is assumed that these factors characterize the assessed process reliably, although it should be noted that, according to other researchers (*Hayton & Allen, 2004*), such a liberal method of factor extraction does not guarantee that the selected factors will not be random. It is therefore suggested to perform parallel factor analysis on a randomly formed data array with the same data structure as the experimental data (*Fabrigar et al., 1999*; *Hayton & Allen, 2004*). Random factors are extracted with parallel factor analysis. When eigenvalues of these random factors are higher or approximately equal to eigenvalues extracted with PCA, the latter values should be rejected as related to random influences. In order to identify non-random factors, we also used parallel factor analysis. In our case the number of non-random factors was four or five, and they explained about 67–75% of the experimental data dispersion.

Next, the analysis of extrema distribution was performed. Such an analysis allows us to reduce the fluctuation of the amplitude of dPT dependence which could mask its periodicity. Analysis of extrema distribution can also reveal whether the periodicity of

a function is stable across sessions. We calculated how many times the changes of the perception of dominant stimulus occurs under each frequency of flashing. Therefore, we calculated the numbers of local extrema (maxima) of $\Delta\tau(k,i)$ (the function of perception time deviation) as a function of the duration of the flashed stimulus:

$$\Delta\tau^1(k,i) = \begin{cases} 1, if\, \Delta\tau(k-1,i) < \Delta\tau(k,i) > \Delta\tau(k+1,i) \\ 0, \text{in other case} \end{cases},$$

i.e., $\Delta\tau^1(k,i)$ will equal 1, if at point $k$ a local maximum of function $\Delta\tau(k,i)$ is observed, otherwise it will equal 0. Next, we summed the number of local extrema over all sessions: $\Delta\tau^1(k) = \sum_i \Delta\tau^1(k,i)$. The value of function $\Delta\tau^1(k)$ at point $k$ is an integer number and defines how many times a local maximum at point $k$ (duration of displayed stimulus) was observed through all sessions. Furthermore, we calculated the mean value $M$ $(\Delta\tau^1(k))$ of function $\Delta\tau^1(k)$. We also calculated how many maxima at point $k$ of function $\Delta\tau^1(k)$ were above and below the value $M$. We assigned "1" for the all values that exceeded the value $M$, and "0" for the all values that were below the value $M$. Thus, we produced a sequence of 1's and 0's. The total number of 1's is named "number of case A", and the total number of 0's is named "number of case B". Thus, we had separate intervals at the $k$-axis filled with 1's and 0's. The number of such intervals is called "number of runs". We used "runs test for randomness" to check whether the distribution of 1's and 0's along the $k$-axis was random or non-random (*Bradley, 1968*).

In a similar way the function $\Delta\tau^0(k)$ describes how the other extrema (minima) points of the function $\Delta\tau(k,i)$ are distributed along the $k$ axis. Because the correlations between functions $\Delta\tau^1(k)$ and $\Delta\tau^0(k)$ were high and equal $-0.8 \div -0.9$, we analyzed only the functions $\Delta\tau^1(k)$.

In order to check whether the obtained functions (factor loadings and function $\Delta\tau^1(k)$) were periodic, we performed periodic function fitting:

$$y(k) = a_0 + A\sin(\pi(k - \varphi_0)/w),$$

in regard to which square deviation of functions was minimal.

## RESULTS

The data revealed that the PT of the dominant image varied across subjects from a few seconds to ten seconds. For example, DOWN was perceived longer than UP by subjects **RB** and **IS** (average perception time for DOWN and UP perception was 2.09 and 1.67 s respectively for **RB,** and 3.65 and 2.57 s for **IS**), but UP was perceived longer than DOWN by subjects **AS** and **AV** (8.4 s *vs* 2.14 s and 4.07 s *vs* 3.04 s). Here small numbers indicate a high alternation rate.

The results of one-way ANOVA revealed the main effect of stimulus flash duration (there were 18 values of flash duration: 0 and 4–20 ms, where 0 = non-flickering condition) for all subjects. The statistical analysis (*post hoc LSD* test) of the experimental data (dPT) confirmed that the differences between the minimum and maximum values were statistically significant (Table 1 and Figs. 2, 3A and 3B).

Vaitkevicius et al. (2018), PeerJ, DOI 10.7717/peerj.6011

**Table 1 ANOVA results for the factor of flash duration for separate subjects.**

| Subject | RB | AS | AV | AS2 | IS | GS | MR | MK |
|---|---|---|---|---|---|---|---|---|
| | $F(17, 32{,}657) = 36.29$; $p < 0.0001$ | $F(17, 5{,}118) = 4.62$; $p < 0.0001$ | $F(17, 8{,}816) = 8.83$; $p < 0.0001$ | $F(17, 9{,}103) = 4.52$, $p < 0{,}0001$ | $F(17, 2{,}945) = 2.64$; $p < 0.0003$ | $F(17, 6{,}807) = 59.31$; $p < 0.0001$ | $F(17, 4{,}766) = 7.62$; $p < 0.0001$ | $F(17, 2{,}208) = 3{,}40$, $p < 0.0001$ |

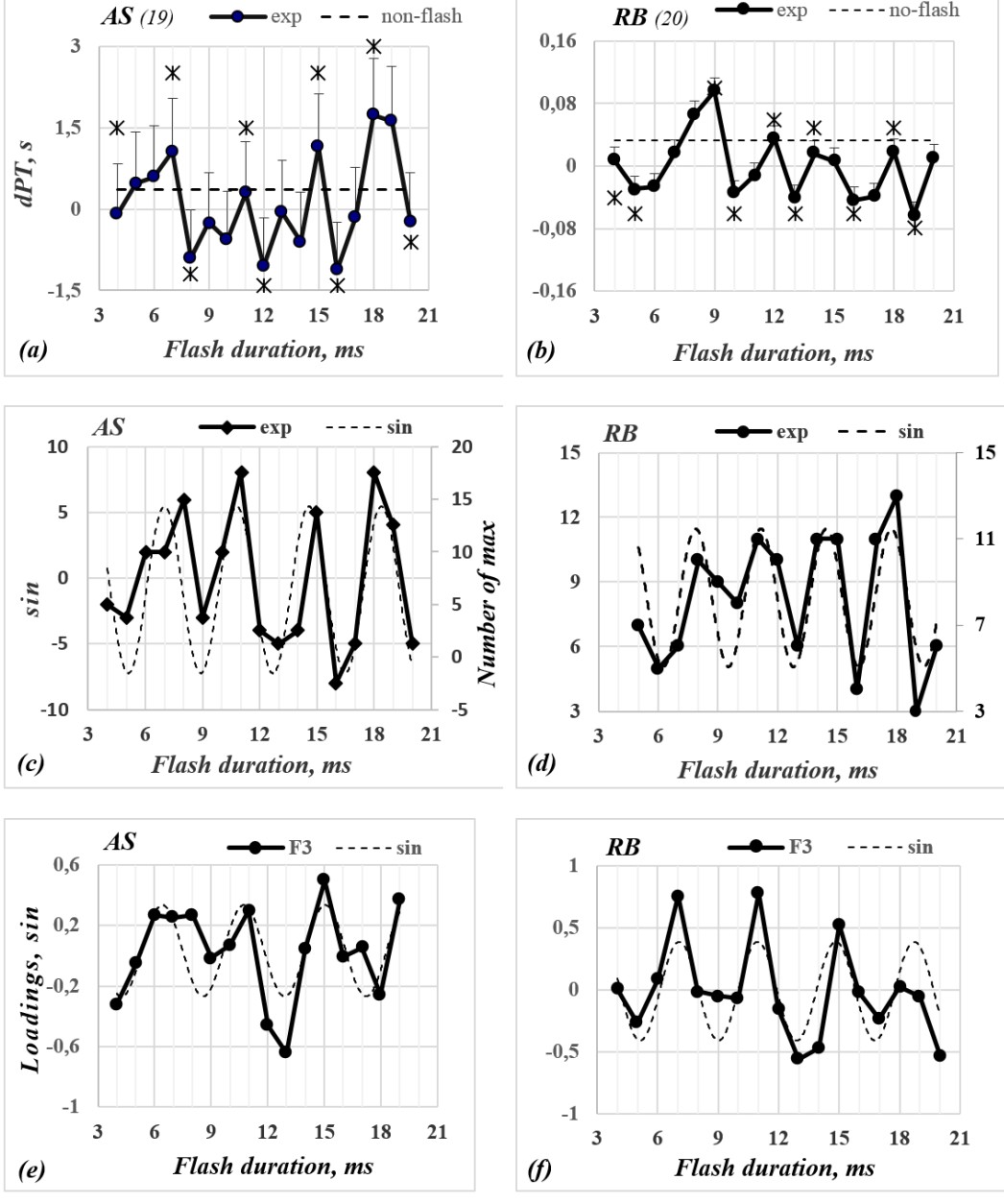

**Figure 2** **The functions of dPT (A, B), the number of local maximum (C, D) and the Factor loadings (E, F) versus the flash duration for observers AS and RB.** (A, B) The abscissae—duration of flashing stimulus (ms), the ordinate—dPT value (s). The continuous curve with filled symbols represents the dPT curves; the dashed line—the dPT of non-flickering stimulus relative to the mean of perception time for all sessions of the given subject. Capital letters on the top of every picture mark different observers (number of all sessions, on which the data was collected, is in the brackets). The points labelled by asterisks mark points, where differences among neighboring extrema of $\Delta\tau(k)$ were statistically significant. (C, D, E, F) The dashed lines are sinusoidal approximation functions.

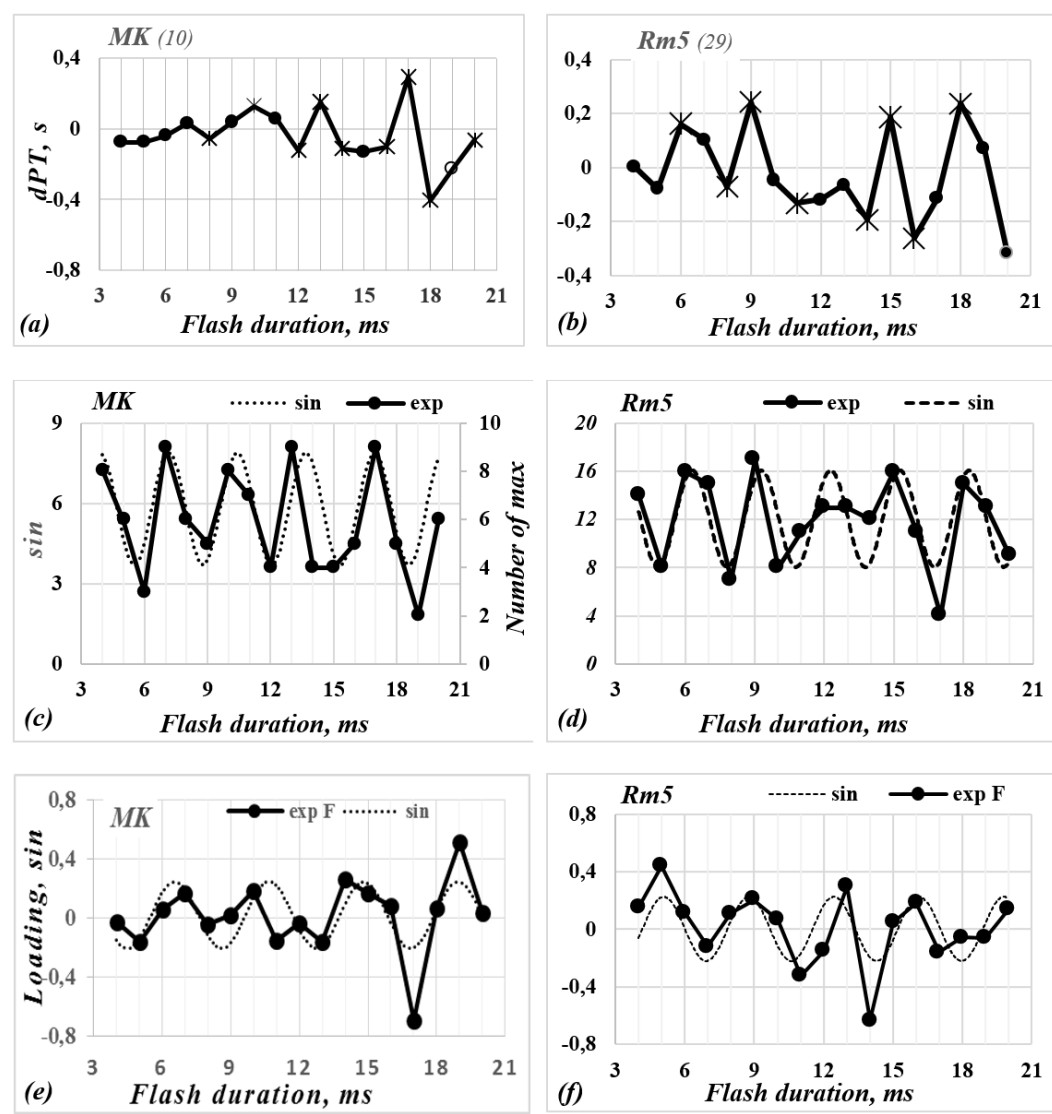

**Figure 3** The functions of dPT (A, B), the number of local maximum (C, D) and the Factor loadings (E, F) versus the flash duration for observers *MK* and *Rm5*. All the symbols are the same as in Fig. 2.

Furthermore, the preliminary statistical analysis leads us to hypothesize that the influence of the stimulus duration could be periodic. In order to check this hypothesis, we searched a periodic functions fitting $y(k)$. Functions $y(k)$ are pictured by dotted lines in Figs. 2, 3C and 3D. Their parameters ($w$) are presented in Table 2. According to the results of the approximation by the sine function, the duration of the stimulus influences the changes of perceived stimulus. This influence repeats periodically. The sine period is $T_{PT} = 2 \times w$, where $w$ equals 3.8, 3.28, 3.42, and 2.98 ms for subjects *AS, RB, MK,* and *Rm5* respectively.

It should be noted that although the $\Delta\tau(k)$ dependences established have a similar periodicity for all subjects, the amplitudes of these functions at the extreme points vary considerably. This could mean that the deviation of perception time (dPT) depends upon

Vaitkevicius et al. (2018), *PeerJ*, DOI 10.7717/peerj.6011

**Table 2  The parameters of $y(k)$ functions.**

| Subjects | AS | | | RB | | | MK | | | Rm5 | | |
|---|---|---|---|---|---|---|---|---|---|---|---|---|
| *Estimated parameter* | $\Delta\tau\,(k)$ | $\Delta\tau^1(k)$ | F | $\Delta\tau\,(k)$ | $\Delta\tau^1(k)$ | F | $\Delta\tau\,(k)$ | $\Delta\tau^1(k)$ | F | $\Delta\tau\,(k)$ | $\Delta\tau^1(k)$ | F |
| $k_1$ | 7 | 8 | 6–8 | 8–9 | 8 | 7 | 7 | 7 | 7 | 6–7 | 6–7 | 5 |
| $k_2$ | 11 | 10–11 | 11 | 12 | 11 | 11 | 10 | 10 | 10 | 9 | 9 | 9 |
| $k_3$ | 15 | 15 | 15 | 14–15 | 14–15 | 15 | 13 | 13 | 14–15 | 15 | 15 | 13 |
| $k_4$ | 18–19 | 18 | 19 | 18 | 19 | 18–19 | 17 | 17 | 19 | 18 | 18 | 16 |
| $T_{PT}$ | Mean = 4 | 3.80, ($w=1.9$) | ~4, ($w=2$) | Mean ≈ 3 | 3.28, ($w=1.64$) | ~4, ($w=2$) | Mean = 3.33 | 3.42, ($w=1.71$) | 4 ($w=2$) | Mean = 4 | 2.98, ($w=1.49$) | 3.68, ($w=1.84$) |
| Average | | 3.93 | | | 3.43 | | | 3.58 | | | 3.55 | |

**Notes.**

$\Delta\tau(k)$, changes of perception time (PT) of dominant image.

$\Delta\tau^1(k)$, number of maximum at point $k$ along the abscissa.

[F] factor loading.

$k_1$, $k_2$, $k_3$ and $k_4$—the location of the first, second, third and fourth extrema peak (maxima) of corresponding function along abscissa ($k$).

several factors, rather than a single factor. That would be in agreement with other authors. For example, it was demonstrated that the so called "stochastic resonance" in the presence of a hypothetical neural noise and "periodic driving" (displaying stimulus) influence the alteration rate of the perception of the dominant image (*Kim, Grabowecky & Suzuki, 2006*). Moreover, according to *Lankheet (2006)*, the adaptation of detectors, and mutual backward lateral inhibition among them, affects the alteration rate of the dominant image. *Pearson & Brascamp (2008)* and *Knapen et al. (2009)* demonstrated that the properties of a so called "perceptual memory" also have an influence on the dominance of the perceived stimulus. Taking these findings into account, a factor analysis (principal component analysis—PCA) was run on the data ($\Delta \tau (k)$).

The PCA identified up to six eigenvectors for each subject. These eigenvectors explain on average 67–75% of the total data distribution. Parallel factor analysis (*Fabrigar et al., 1999*; *Hayton & Allen, 2004*) was applied to identify non-random (significant) factors. As a result, it can be argued that four or five factors are non-random. Another purpose of the PCA was to find whether there are factors that determine the periodicity of obtained dependence. The analysis of non-random factors revealed that: (i) there is one factor ($F_3$) that exhibits the periodicity; (ii) the periodicity of this factor is similar to obtained dPT periodicity; (iii) the periodicity of this factor is similar for different subjects. The values of $F_3$ factor loadings versus the duration of stimulus are shown in Figs. 2, 3E and 3F).

The dPT and maxima functions are approximately periodic (Figs. 2, 3A, 3B, 3C and 3D). The standard peak analyzer procedure (as in *OriginPro 9.1*) was used to determine the maxima of functions. The locations of peaks for the different functions varied slightly. The first ($k_1$), second ($k_2$), third ($k_3$), and fourth ($k_4$) peaks are located along the abscissa axis on intervals (5–7), (9–11), (13–15), and (16–19) ms respectively. Differences in locations of the peaks for **RB**, **AS**, **MK** and **Rm5** are approximately equal to 2 ms. However, the factor loadings have four or five peaks, which are repeated at about the same value of ∼3.5–4.0 ms (see Figs. 2 and 3 and Table 2). That means the obtained functions are shifted in phase relative to each other and their periods differ slightly (from 3.68 to 4.00 ms).

It should be noted that for three observers (**RB**, **AL** and **MK**) the location of the peaks of factor loadings approximately coincide with the peaks of dPT functions and functions of maximum number (Figs. 2E, 2F, 3E and 3F). The first maxima are located at about the same interval ∼7–8 ms. The second, third and fourth peaks are at 10 ∼11, ∼13–15, and ∼17–19 ms respectively. The averaged distance among the peaks of all three functions for three observers (**RB**, **AS**, **MK**) are 3.93, 3.43, and 3.58 ms respectively. All functions for **Rm5** data are also periodic, although their periods are slightly shorter. The averaged periods of $\Delta \tau (k)$, $\Delta \tau^1(k)$, and $F_3$ equal 4, 2.98, and 3.68 ms respectively.

Some years ago, we addressed the problem of how the duration and frequency of flashing binocular competitive images affect the unstable perception in the case of binocular rivalry. It was shown that while the flash frequency was changing, the rate of perceptual alteration varied periodically (*Geissler et al., 2012*). In order to examine whether the PCA results obtained in case of the perception of ambiguous figures and binocular rivalry are similar, we compared these results. Figure 4 graphically presents the factor loadings for experimental data on binocular rivalry and on Necker cube perception.

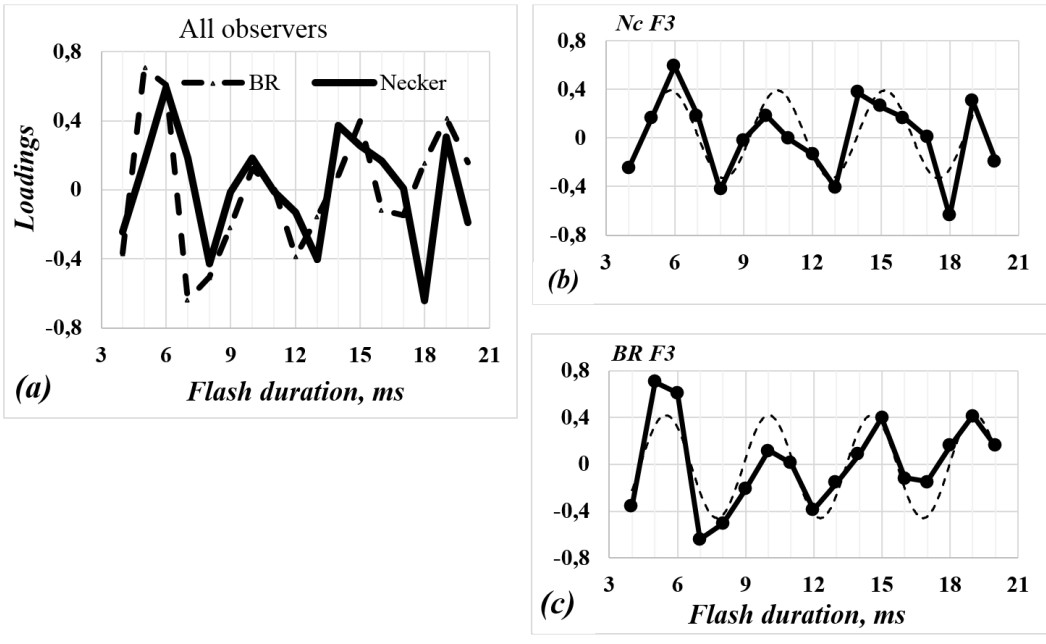

**Figure 4 Factor (F3) loadings against duration of flashing stimulus.** (A) The continuous solid and dashed point-like curves are the factor loadings (F3) calculated for the Necker cube and binocular rivalry data of two different groups of observers, respectively (*Geissler et al., 2012*). (B, C) The solid line indicates factor loadings for Necker cube (*Nc*) and binocular rivalry (*BR*), respectively. The dashed lines show sine function approximation.

Factor loadings obtained on the aggregated data for all subjects who participated in the Necker cube and binocular rivalry experiments are approximated by the following functions:

$$y_{Nc}(k) = 0.03 + 0.36 sin[\frac{\pi(k+0.02)}{2.34}] \quad (F = 5.24; p = 0.02) \text{ and}$$

$$y_{Br}(k) = -0.02 + 0.44 sin\left[\frac{\pi(k+0.19)}{2.27}\right] \quad (F = 6.67; p = 0.006) \text{ respectively.}$$

In both cases the period of sine function is similar (equals 4.68 and 4.54 ms respectively). The periods of experimental curves equal 3.5–4 and 4–5 ms respectively, i.e., the differences are rather small.

## DISCUSSION

Analysis of the experimental results demonstrate that the frequency of a flashing stimulus influences the perception time of a dominant image, extending or shortening it. Furthermore, the influence of the stimulus frequency on the alternation rate of perception was periodic with 4 ms intervals. The extrema (maxima or minima) of all three functions (perception time $\Delta\tau(k)$, number of maxima $\Delta\tau^1(k)$, and loadings on F$_3$ factor) recur along the $k$-axis not only periodically, but also are located approximately at the same positions along the $k$ axis (see Figs. 2 and 3 and Table 2). A similar analysis has been conducted by *Fesi & Mendola (2015)*, who found an inverse correlation between the alternation rate and the peak frequency of late evoked gamma activity in the primary visual cortex (in regions

V1 and V2) for bistable images. How could we explain that the probability of change of perception versus the flicker frequency (or duration) of stimulus is a periodic function?

## Hypothetical mechanism of interaction between internal rhythm and sequence of external stimulus presentation

Considering the influence of a rhythmically flickering stimulus on the alternation rate of a dominant image, it should be noted that the period of flicker is about $2\times4$ ms $= 8$ ms. In other words, we assume that there is some internal rhythm, which specifies the discrete shortest time moments, when the sensory system input is the most sensitive. If the frequency of the stimulus presentation is a multiple of the frequency of this internal oscillator, then the efficiency of the stimuli should recur and be maximum every 8 ms. Thus, according to these results, the frequency of an internal oscillator should be approximately equal to $10^3/(2\times 4.0) = 125$ cycles/s which is within the high-gamma frequency brain rhythms (80–200 Hz, see *Crone, Sinai & Korzeniewska, 2006*). We cannot speculate what exactly the nature of this internal rhythm is, but we suggest that it should be related to a high-gamma brain rhythm (neural oscillations) that are investigated by various scientists in electrophysiological and neurophysiological studies. There are many brain rhythms of different frequencies that are related to different perceptual, memory related, or other cognitive functions. High-gamma rhythms were observed during visual perception tasks (*Lachaux et al., 2005*), attention tasks (*Ray et al., 2008*), language (*Korzeniewska et al., 2011*), and other cognitive processes. Correlation of brain rhythms with perception suppose the idea that perception (and other cognitive processes) is a discrete process (*VanRullen & Koch, 2003*).

If a sequence of input stimuli coincides with a sequence of electrical activity of some internal oscillator, the time span of a stimulus presentation completely overlaps the time span when the system is maximally susceptible. In this case, we can speak about synchronization of a sequence of external stimulus with a rhythm generated by an internal generator. This agrees with the experimental data of other authors (*Vanagas et al., 1976*; *Geissler, 1987*; *Vanagas, 2001*; *Hasenstaub et al., 2005*; *Geissler et al., 2012*; *Fesi & Mendola, 2015*).

While the duration of flashing stimulus and flashing frequency were entangled in our experimental paradigm, with the duration of stimulus as half of total period, we suppose that the frequency could be a determinant factor for periodicity of the dependence of perception time on flashing stimulus duration. For example, lengthening the optimal duration of stimulus by 1, 2 and 3 ms, its efficiency initially reduces and recovers only after lengthening it by 4 ms. After this its efficiency reduces again, until it is lengthened again by another 4 ms, and so on. Thus, the duration of displaying stimulus alone does not determine the observed effect—the efficiency of stimulus varies every 4 ms, i.e., it is also related to the moment of time when the stimulus is switched on. Thus, it can be related to the accurate coincidence in time of two streams (external and internal) of neuron impulses (*Huber et al., 2008*; *Stanley et al., 2012*). That means that the precise timing of the switching on of the stimulus and action of the internal impulses is important in order to produce the optimal influence on the alternation rate of perceived stimulus.

The question is whether such an information processing method, when the static signal is differentiated with respect to time, can occur under natural conditions, when there is no flashing signal. An investigation of neuronal processes in the retina (*Roska, Molnar & Werblin, 2006*; *Hsueh, Molnar & Werblin, 2008*) confirmed that the differentiation of signals with respect to time could be initiated at the low level of the visual system. It is shown that ganglion cells receive excitation signals from bipolar cells and inhibition delayed signals from amacrine cells. Due to this interaction, ganglion cells get differentiated signals with respect to time. Thus, at the outputs of ganglion cells, a high-frequency sequence of discrete signals can be formed (*Vaitkevicius et al., 1983*). Moreover, it is well known that the eye is constantly moving, hence the image of an object is shifted in time from one place to another on the retina. Amplitudes of the small movements (or ocular tremor) are about 20–40 arcsec with a frequency of ∼90 Hz (*Martinez-Conde, Macknik & Hubel, 2004*), but in extreme case it can go up to 150 Hz (*Spauschus et al., 1999*; *Carpenter, 1988*). There are also experimental findings confirming that the micro movements of the eyes could be involved in this low-level coding process of sensory information (differencing of signals with respect to time and space) (*Kulikowski, 1971*; *Leopold & Logothetis, 1998*; *Roska, Molnar & Werblin, 2006*).

An alternative explanation of our findings, which does not related to the process of synchronization of two streams, could be based on a perceived brightness of stimulus. It is well known that perceived brightness of flashing stimulus depends on the frequency of flashing and stimulus duration (Talbot-Plateau law). In other words, the perceived brightness depends on stimulus power (which in our case is constant). The law holds when the frequency of flashing stimulus is higher than the critical flicker fusion (*Hecht & Wolf, 1932*; *Bartley, 1938*; *Bartley, 1939*); otherwise, the influence of rhythmic stimulus on perception is more complex. In any case, the dependence of stimulus brightness on stimulus duration and flash frequencies used in our study should be more monotonic and cannot explain the periodicity observed in our experiments.

## A common mechanism for binocular rivalry and perception of ambiguous figures

It is also important to note that according to our data, the influence of the frequency of stimulus flicker on the alternation rate of the dominant image perception is similar in both phenomena of bistable perceptions: binocular rivalry and monocular perception of bistable images. Binocular rivalry originates from the different images presented to the retinas of each eye: it is impossible for the human to perceive two different stimuli at the same point in space. At any given moment of time only one object (the dominant image) is perceived, and another object (the image on the retina of the other eye) is not perceived—it is suppressed. The situation is different in the case of the Necker cube: the image of a cube is displayed in the retina of one eye. In other words, two different Necker cubes can create exactly the same perceived image. Since it is impossible to perceive two different objects at the same time and in the same point in space, the subject perceives only one of two possible images (the dominant image) at different moments in time, and any other possible perceptual option is suppressed. The loadings on the factors (F3) as a function of stimulus

duration are similar both in the case of binocular rivalry and in the case of the Necker cube (see Fig. 4). Thus, we can assume that these factor loadings are the result of similar processes involved both in monocular and binocular perception. Comparing perception of ambiguous figures and binocular rivalry, *O'Shea et al. (2009)* previously drew the same conclusion. On the other hand, (*Cao et al., 2018*) argue that experimental data suggest partially independent processes for bistable perception of different types of stimuli.

## CONCLUSIONS

Our paper addresses the problem of how the flickering image of a Necker cube influences the alternation rate of the perception of an ambiguous figure. We measured the durations of the perception of a dominant stimulus and calculated the changes in the duration of the dominant stimulus perception versus the frequency and duration of a displayed Necker cube. The obtained functions of perception time, factor loadings, and number of maxima demonstrate that the alternation rate of a Necker cube changes periodically as a function of flashing stimulus frequency. Maximum effect of the frequency of flashing stimulus on the duration of the perception of a dominant image recurs periodically at approximately 8 ms intervals which suppose the existence of internal rhythm of 125 cycles/s for bistable visual perception.

### Funding
This work was supported by grant MIP-015/2011 and by grant MIP-013/2012 from the Research Council of Lithuania. The funders had no role in study design, data collection and analysis, decision to publish, or preparation of the manuscript.

### Grant Disclosures
The following grant information was disclosed by the authors:
The Research Council of Lithuania: MIP-015/2011, MIP-013/2012.

### Competing Interests
The authors declare there are no competing interests.

### Author Contributions
- Henrikas Vaitkevicius conceived and designed the experiments, analyzed the data, contributed reagents/materials/analysis tools, prepared figures and/or tables, authored or reviewed drafts of the paper, approved the final draft.
- Vygandas Vanagas conceived and designed the experiments, analyzed the data, authored or reviewed drafts of the paper.
- Alvydas Soliunas performed the experiments, analyzed the data, prepared figures and/or tables, authored or reviewed drafts of the paper.
- Algimantas Svegzda conceived and designed the experiments, performed the experiments, analyzed the data, contributed reagents/materials/analysis tools.

- Remigijus Bliumas performed the experiments, authored or reviewed drafts of the paper.
- Rytis Stanikunas performed the experiments, contributed reagents/materials/analysis tools, prepared figures and/or tables, authored or reviewed drafts of the paper.
- Janus J. Kulikowski analyzed the data, authored or reviewed drafts of the paper.

## Human Ethics

The following information was supplied relating to ethical approvals (i.e., approving body and any reference numbers):

The Vilnius Region Ethics Commitee of Biomedical Research granted approval to carry out this study (158200-13-578-173).

## Data Availability

The raw data are provided in a Supplemental File.

## Supplemental Information

Supplemental information for this article can be found online at http://dx.doi.org/10.7717/peerj.6011#supplemental-information.

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
