# Peer review of "Fast cyclic stimulus flashing modulates perception of bi-stable figure"

_PeerJ, doi:10.7717/peerj.6011_

## Round 0.1 · original submission · Major Revisions

Two reviewers have read your study and requested extensive revisions. Please address each concern line-by-line in your resubmission.

·

Basic reporting

This manuscript is not well structured and very difficult to read. It took me some time to understand the aim and results of the research: basically, participants were shown a bi-stable figure (figure that can be interpreted in two distinct ways) that was flashed on a screen by rhythmically turning it on and off; perceptual experience (rate at which percepts alternate and time a percept is sustained) was then measured as a function of flash duration and frequency. The main conclusion was that the alternation rate was modulated by the flash duration.
The authors failed to write an introduction that invites the reader to be drawn into their research: it lacks a clear hypothesis and does not explain why their research is important; instead they provide an incoherent overview featuring feature detection and brain waves that was confusing rather than clarifying.
Throughout the manuscript the authors refer to "an interaction between two streams: an external one (controlled and caused by the flash frequency and duration) and an internal one"; what that internal stream should look like, or how it is generated, is very unclear.

Experimental design

Just like the rest of the paper, the methods that were used were very unclear and hard to understand. The authors could of have used visuals or explain the concept better; for instance, they define absolute perception time (PT), then follow by the statement: “instead of PT we measured the changes of PT”; instead of explaining what this means conceptually, they immediately proceed with a confusing formula. Because the authors fail to explain the methods in a comprehensive fashion, I perceived the methods as abstract and detached from the experiment itself.

Validity of the findings

Given the lack in clarity and detail of the methods that were used (see 2), it was hard to judge the validity of the finding. However, based upon the p-values reported in table 1, and upon visual inspection of the figures 1-3, the claim that modulation of perception as a function of flash duration, is periodically at 4 ms intervals, seems to be quite robust.

Additional comments

Dear authors: I recommend seeking the help of an experienced writer. I struggled to understand your manuscript and was confused from line one until the very end. The manuscript lacks clarity and consistency and is incomprehensible in its current form. Here are some questions/doubts/comments.

lines 2-3 (abstract): “We investigated whether the alternation rate of the perception of a Necker cube depended on the degree of synchronization between two streams of spikes”. Sorry, that is not what you did!

Lines 7-9 (abstract): “Knowing how a flickering stimulus with a given frequency and duration affects the alternation rate of bi-stable perception we could estimate properties of the
internal signal”; if true, then what are the properties and why are they not listed or mentioned in the abstract?

Lines 11-12 (abstract): “The values of the stimuli…”. I have no idea what “value” means in this context;

Introduction: throughout the manuscript you bring up that mysterious “internal signal”, which you referred as spikes and internal stimuli (abstract); or as “internal high frequency rhythms” and “internal impulse streams” (introduction). It is all very confusing. I recommend rewriting the introduction and focus on 3 points: (1) Explain what made you investigate how perception can be modulated by flash duration and frequency. In other words, why was it worth doing this effort? (2) Clearly state your hypothesis (3) Provide arguments for your hypothesis. For instance, what supports the existence of an internal stream that is independent of external influences and oscillates at a frequency that is sufficiently high (125 or 250 Hz) to explain your results.


Methods: the methods could use a figure for clarity. For instance: panel A, a cartoon of the setup; panel B: an image of the exact version of the Necker cube used for the experiment (with an arrow pointing from panel B to the location in panel A where the Necker cube is shown in the setup. Panel C: an example of a session, with 19 lines corresponding to the 19 blocks; each line represents 180 seconds and portions of the line are colored in one of two colors that represent when the subject was perceiving UP or DOWN. Then you can refer to this figure to explain the mathematical methods (for instance absolute perception time can then be explained as the total time represented by color 1, perceptual changes explained as … etc.)

Line 122: “... the mean dPT of UP and DOWN” should be “... the mean dPT of respectively UP and DOWN”

Data analysis: should be written more concise; in addition to explain the methods mathematical, an effort should be done to explain each method conceptually as well.

Line 146-147: It is not clear why 5 subjects were lumped together. This basically reduces the data to 4 sets: 3 from experienced subjects and 1 averaged across all naïve subjects; the results are thus heavily skewed in favor of subjects that are not naïve, presumably the authors of the manuscript; some concerns of bias might arise.

Lines 166-167: “The data revealed that the PT of the dominant image varied in timing from a few seconds to ten seconds depending on the subject.” Varied with respect to what? The non-dominant image, varied across blocks? Varied across sessions?

Lines 167-169: “For example, DOWN was perceived longer than UP by subjects RB and IS (2.09 s vs 1.67 s and 3.65 s vs 2.57 s respectively), but …” what does the 2.09 vs 1.67 represent? It is my understanding that each block last 180 s and that there are 19 blocks in a session (including the non-flashing condition); thus totaling 180 s * 19 = 3420 seconds in a session; this means that the time DOWN was perceived + the time UP was perceived should be 3420 seconds. I do not understand how the numbers mentioned in lines 167-169 are obtained. Please explain.

Result section: Could be explained more concise; many details are not relevant

Discussion section: reads like a fishing expedition, desperately trying to explain the “internal stream”; based upon your results, the opposite is true: oculomotor tremor (80-100 Hz) does not fit because the frequency range does not match; the non-linearity of the Broca-Sulzer effect is observed around 80-100 ms. I am not sure how this can be applicable to your findings. The gamma range (40-90 Hz) is also does not fit. So I think it would be better to organize the discussion as follow: we checked oculomotor tremor as a potential generator of our internal signal, but need to rule this out because of this and that; we checked gamma oscillations, but need to rule it out because… etc..

The discussion should also be used to consider alternative explanations for the results, not based upon an “internal stream”; eye-blinks for instance induce reversals in perception, and the rate of eye-blinks might be modulated by the flash duration and frequency.

·

Basic reporting

The article is in general well structured, but it would be clearer if the description of the sinusoidal fitting were moved to the Methods section.
The authors should also clarify if the results described in lines 170-175 are calculated over dPT or deltaTao.

Experimental design

The authors performed the experiment on 8 participants, 3 non-naïve and five naïve. The amount of data collected on the different subject varies significantly, with some subjects taking part in 3 sessions, some in 10, and some in 20. An explanation of this discrepancy should be present in the text, and potentially the analysis should be repeated in such a way that there is the same amount of data for all subjects (i.e. the first three sessions for all subjects).

Validity of the findings

The major flaw I find in the manuscript is that the final data analysis presented is performed on heavily processed data, including two non-linear transformations. At the very least, a representation of the raw data, the changes in PT or dPT as a function of flash duration, should be provided along with an explanation of why this processing was performed and why it is the best way to analyze the data. As it is, it is hard to see if the conclusions are supported by the data or if they are a consequence of the performed manipulations.
Other issues I found with the data analysis and interpretation are as follows:
1. The PCA analysis is not very clear. What is the dimensionality of the inputs to the PCA, and what are these inputs? The PCA was also run on the three of the subjects individually, since those are the ones that are claimed to perform 20 trials (although only two of them participated in three sessions as described previously, so this should be clarified), and on the aggregated of the other five subjects. Why is this? It seems like the correct way to do this would be to run the PCA on the aggregate of the 8 subjects if it was not possible to do it individually for each subject.
2. Regarding the results of the PCA, it is unclear what insight this PCA analysis provides. It is specified that loadings F3 or F4 depended on stimulus duration. Assuming that subindices 3 and 4 specify the order of their weight (i.e., the third and fourth-most important factors), it would be interesting to know: a) what other factor affected these loadings, and b) what parameters were the most relevant (those that affect F1 and F2).
3. In lines 303-315 the authors discuss the effect of stimulus duration in perceived brightness and cite the presence of the Broca-Sulzer effect at specific durations, but it is unclear how this discussion relates to the data at hand.
4. In general, it feels the discussion could be made more concise.

Additional comments

There are two points that I would like to be included in the discussion:
1. Whereas the authors claim the data point toward the existence of an internal oscillator of frequency around 125 Hz, it seems like the data could be explained with other models that do not include this type of oscillation. The authors even hint towards a potential role of microscopic eye movements (tremor). Why do the authors think the internal oscillator is the most parsimonious explanation?
2. How would a change in stimulus duration would affect the percept time of the bistable stimulus if flashing frequency was to be kept constant (i.e. 1 ms flash on an 8ms period versus a 6 ms flash with 8 ms period), depending on the model chosen?

---

## Round 0.2 · Major Revisions

Two reviewers have read your manuscript and feel that there has been significant progress (thank you!), but also feel that there are substantive improvements yet required. They have made specific suggestions--which you should implement--and also have clearly outlined remaining ambiguities. Please fix these. I'm optimistic that the paper will eventually be publishable, and I would be grateful if you would put in the effort now to follow the latest suggestions as fully as possible so that we needn't have more than one more cycle of review.

·

Basic reporting

See General comments

Experimental design

See General comments

Validity of the findings

See General comments

Additional comments

I am impressed with the improvements made by the authors. My previous comments are addressed in a satisfactory manner. However, the language throughout the manuscript is often too verbose, and the many grammar and spelling errors give a sloppy impression; I tried to catch some errors (see comments) but I am sure that I missed quiet a few; I recommend seeking the help of a native English speaker to “polish” the paper;

Comment 01: bistable (lines 30, 73, 75, 326 and 341) or bi-stable (title, lines 8, 13, 36 and 270)? Please be consistent.

Comment 02: lines 9-11 “… show that THE DURATION of the dominant stimulus perception depends …”
lines 13-14 “We can also conclude that it is NOT THE STIMULATION DURATION but the ….”
Please clarify this apparent contradiction.

Comment 03: lines 12-13: “… could be explained by existing of internal rhythm …” Suggested: “… could be explained by the existence of an internal rhythm”

Comment 04: line 61 “can not”; preferred: “cannot”

Comment 05: l “synchronisation” (line 62) or “synchronization” (lines 6 and 286)? Please be consistent.

Comment 06: line 63: “rythmically” should be “rhythmically”

Comment 07: line 67: “stumulation” should be “stimulation”

Comment 08: line 69: “in the study”; suggested: “in a study”

Comment 09: line 73: “is continuation”; suggested: “is a continuation”

Comment 10: line 110: “later in the work is named as”; suggested: “which we refer to as”

Comment 11: line 111: “minimal flash duration”; suggested: “minimum flash duration”

Comment 12: line 119: if PT represents the time accumulated across a block, then please add this information.

Comment 13: line 124: “Firsly” should be “Firstly”

Comment 14: line 158: “non-random factor” should be “non-random factors”

Comment 15: line 165: “…will repeat again, will increase, if the frequency…” I did not understand this sentence; please check use of commas or reformulate.

Comment 16: line 165-166: “Thus to avoid…” should be “Thus, to avoid…”

Comment 17: line 173: “… integer number, and defines…”; suggested: “integer number and defines…”

Comment 18: lines 195-197: For clarity, I suggest mentioning (explicitly) that small numbers indicate a high alternation rate.

Comment 19: please compare lines 242 and 244: in an enumeration, you sometimes use units for each value; sometimes none at all; I suggest that you, throughout the paper, use units only when mentioning the last value: for instance: 10, 20 and 30 s.

Comment 20: line 245: “Some years ago we…” should be “Some years ago, we… “

Comment 21: line 256: I am confused by this sentence; because you start the sentence with "in both cases...", I expect a single value (for instance in both cases the period equals X).

Comment 22: line 262-263: “…duration of flashing stimulus influence the perception time…” should either be: "... duration of a flashing stimulus influences the perception time..." or "duration of flashing stimuli influence the perception time…"

Comment 23: line 276: “Considering the influence of rhythmically flickering stimulus on…” should either be “Considering the influence of a rhythmically flickering stimulus on…” or “Considering the influence of rhythmically flickering stimuli on… ”.

Comment 24: line 283: “… to high-gamma frequency brain rythms…” should be “…to high-gamma frequency brain rhythms…”

Comment 25: line 290: “It should be noted that our results allow to propose that…”: too verbose

Comment 26: line 296: “Thus it can be related…” should be “Thus, it can be related…”

Comment 27: line 309: “…(or tremor)”; suggested “…(or ocular tremor)”

Comment 28: line 310: most investigators in fixational eye movements provide lower estimates for the frequency of ocular tremor; please revise.

Comment 29: lines 319-320: since “Critical Flicker Fusion” in only used once, I don’t see the need to provide the abbreviation.

Comment 30: line 350: “…maxima demonsrate that…” should be “…maxima demonstrate that…”

Comment 31: line 353: “…suppose the existance of…” should be “…suppose the existence of…”

·

Basic reporting

no comment

Experimental design

After the rewriting of the methods section, I still have some issues with the methods as described:

-I still do not understand what the PCA analysis adds on top of the rest of the data. It seems to me like components above the noise level were extracted, and then the one component that matched the experimenters hypothesis, which was not even the same for all subjects nor the one with the highest weight, was represented. I suggest authors either clarify exactly what they were trying to achieve with this analysis and the procedure, and provide a more extensive discussion of the results, or remove the analysis altogether.

- The description of the analysis methodology still needs some improvement. As it stands, it is very hard to understand and requires to go back and forth paragraphs to understand the analysis of extrema distribution in particular. A more descriptive naming of variables would be a start.

- I would like to see the breakdown of the data for the group of subjects labeled as Rm5, even if the PCA analysis is not possible.

Validity of the findings

Several issues here. In the results section:

-"The statistical analysis (post hoc LSD test) of the experimental data (dPT) confirmed that the differences between the minimum and maximum values were statistically significant, and the extreme points were recurrent": what does it mean for the extreme points to be recurring? How was this assessed? Where are the results for the LSD analysis? I only see the ones for the basic ANOVA

Regarding the discussion and conclusions: the authors need to explain better what this internal oscillator they propose is exactly. Are they talking about an internal rhythm generated somewhere in the brain and then distributed across different areas? Where is it coming from? Or is it a matter of input timing as it relates to processes internal to the cells? Something else? The suggested explanations are extremely vague and highly speculative. The fact that the periodicity they observe falls within the bandwidth of ocular micro tremor and high-gamma rhythms seems irrelevant given the information provided here, since it can easily be a coincidence (specially with the big bandwidth associated with both phenomena, which makes this type of coincidence easier).

Also, the authors claim that these results show that only frequency of flashing and not flash duration influence bistable perception, which is not true. The effect of these parameters can not be disentangled from a duration effect by the fact that in their experimental paradigm those two variables follow a one-to-one relationship (the authors themselves use duration to refer to frequency at some points in the manuscript). Whereas the data points toward flashing frequency as a relevant factor, effects of duration can not be discarded.

It is unclear what the authors are trying to communicate in the paragraph explaining the effect of duration, flicker frequency, and perceived contrast.

---

## Round 0.3 · Minor Revisions

Both reviewers are satisfied that the manuscript is much improved. They have a few small further improvements. Please make these adjustments and resubmit.

·

Basic reporting

No comment

Experimental design

No comment

Validity of the findings

No comment

Additional comments

The manuscript underwent, after two rounds of editing, a major transformation; given that the authors can resolve some minor issues (see below), I can now finally recommend the publication of the manuscript in PeerJ.

Some minor comments:

Line 25: “… time of dominant stimulus depends on...” should be either “… time of the dominant stimulus depends on“ OR “… time of dominant stimuli depend on…”

Line 28: “125 cps”; but line 295: “125 c/s ”. Since neither “cps” nor “c/s” is defined, maybe use “cycles/s” in both cases?

Line 44: “… physical properties of stimulus… “ should be replaced either by “… physical properties of stimuli… “ OR “… physical properties of a stimulus…”

Line 58: “… the absence of stimulus on …” should be replaced either by “… the absence of stimuli on …” OR “… the absence of a stimulus on …”

Line 175: “… about 67 – 75 % …”; but line 236: “average 66.72– 75%...”; begin and end of the range should show the same precision, or both should be rounded.

Line 249: “…(from 3.68 to 4 ms)…”: same comment as above; if the 4 ms has a precision of 3 digits, then use: “…(from 3.68 to 4.00 ms)…”.

·

Basic reporting

no comment

Experimental design

The explanation and interpretation of the PCA analysis is much clearer in the rebuttal letter than in the manuscript. Please rewrite using that as a model.

Validity of the findings

Make the fact that the effect of perceived brightness of stimulus due to temporal effects does not affect the periodicity (due to its monotonic shape in the range of durations and frequencies tested) clearer.

---

## Round 0.4 · accepted · Accept

Thanks you for making your final revisions to this paper. I will recommend to publish!

#